# Area-selective atomic layer deposition on 2D monolayer lateral superlattices

Jeongwon Park [1,12], Seung Jae Kwak [2,12], Sumin Kang[1,12], Saeyoung Oh[3], Bongki Shin[4], Gichang Noh [1,5], Tae Soo Kim[1], Changhwan Kim[6], Hyeonbin Park[1,7], Seung Hoon Oh[7], Woojin Kang[2], Namwook Hur[6], Hyun-Jun Chai[1], Minsoo Kang[1], Seongdae Kwon[1], Jaehyun Lee[1], Yongjoon Lee[8], Eoram Moon[1], Chuqiao Shi [4], Jun Lou [4], Won Bo Lee [2], Joon Young Kwak [9], Heejun Yang [8,10], Taek-Mo Chung[7], Taeyong Eom [7], Joonki Suh [3,6], Yimo Han [4], Hu Young Jeong [3], YongJoo Kim [11] ✉ & Kibum Kang [1,8] ✉

The advanced patterning process is the basis of integration technology to realize the development of next-generation high-speed, low-power consumption devices. Recently, area-selective atomic layer deposition (AS-ALD), which allows the direct deposition of target materials on the desired area using a deposition barrier, has emerged as an alternative patterning process. However, the AS-ALD process remains challenging to use for the improvement of patterning resolution and selectivity. In this study, we report a superlattice-based AS-ALD (SAS-ALD) process using a two-dimensional (2D) $MoS_2$-$MoSe_2$ lateral superlattice as a pre-defining template. We achieved a minimum half pitch size of a sub-10 nm scale for the resulting AS-ALD on the 2D superlattice template by controlling the duration time of chemical vapor deposition (CVD) precursors. SAS-ALD introduces a mechanism that enables selectivity through the adsorption and diffusion processes of ALD precursors, distinctly different from conventional AS-ALD method. This technique facilitates selective deposition even on small pattern sizes and is compatible with the use of highly reactive precursors like trimethyl aluminum. Moreover, it allows for the selective deposition of a variety of materials, including $Al_2O_3$, $HfO_2$, Ru, Te, and $Sb_2Se_3$.

Area-selective atomic layer deposition (AS-ALD) induces material deposition on the desired area by pre-defining a surface with different chemical activities[1–5]. The general procedure in preparing the pre-defined template for the conventional AS-ALD (CAS-ALD) is patterning a chemically inert barrier material using a top-down approach, which can prevent the chemisorption and subsequent reaction of ALD precursors on the substrates, leading to selective deposition on areas in which the barrier is absent[1–3] (see Supplementary Note 1 for more details).

Interestingly, we observed AS-ALD on a two-dimensional (2D) $MoS_2$-$MoSe_2$ lateral superlattice as a patterning template. The 2D superlattice-based AS-ALD (SAS-ALD) is fundamentally different from the CAS-ALD. First, the length scale of the pattern fabricated by SAS-ALD can potentially overcome the fundamental resolution limit in the CAS-ALD. In the growth of 2D van der Waals materials, such as $MoS_2$ and graphene, lateral growth can occur as a result of the nature of the crystal structure[6–10]. With this phenomenon, the lateral superlattice of 2D transition metal dichalcogenides (TMDs) has been demonstrated by supplying the precursors sequentially in the chemical vapor deposition (CVD) process[11–13], in which the width of superlattice can be controlled precisely with duration time of the precursors. Therefore, the 2D superlattice offers an ultra-high-resolution pre-defined

template for selective deposition, and the resulting pattern is not limited to optical diffraction and can be improved further to the atomic scale. Second, the mechanism of SAS-ALD is distinct from that of CAS-ALD. While CAS-ALD is mainly governed by chemisorption and reaction[1–4], physisorption of precursors on the surface of the barrier material is hard to completely avoid, as some undesired depositions happen due to intermolecular interactions such as London, Debye, and Keesom forces[1–3,14]. Furthermore, small and reactive precursors such as trimethyl aluminum (TMA) can penetrate barrier materials, making it difficult to achieve highly selective deposition[15–20]. By contrast, in SAS-ALD, chemical reaction between the precursor and substrate is prevented on both surfaces of $MoS_2$ (blocking area) and $MoSe_2$ (deposition area); this is due to the crystal structure of $MoS_2$ and $MoSe_2$, in which the chemically unsaturated dangling bond does not exist, thus providing a chemically inert surface for ALD[14,21–24]. Instead of chemisorption, the selective deposition on the 2D superlattice originates from the physisorption and diffusion of ALD precursors. Therefore, in SAS-ALD, even when using highly reactive ALD precursors, remarkable selectivity can be achieved, and selective deposition is valid in very narrow patterns. In this report, we analyzed the mechanism of SAS-ALD which is mainly caused by physisorption difference and diffusion of precursors.

## Results

### Superlattice-based AS-ALD

Figure 1 shows the entire process of SAS-ALD, which consists of a CVD step for lateral superlattice growth (fabrication of the AS-ALD template) and a following ALD step for selective deposition on the superlattice. Figure 1a schematically explains the growth of a monolayer $MoS_2$ (yellow)-$MoSe_2$ (red) lateral superlattice through the CVD process, supplying diethyl sulfide (DES) and dimethyl selenide (DMSe) alternatively in gas-phase (see Methods for the details). Figure 1b shows a representative scanning electron microscopy (SEM) image of

the lateral superlattice, which has the $MoS_2$ region (dark gray) and the $MoSe_2$ region (light gray). By controlling the flow of gas-phase precursors, we can make a superlattice with varying pitch sizes within a single flake (Supplementary Fig. 1a–c). Each $MoS_2$ and $MoSe_2$ region can have widths ranging from 10 nm to hundreds of nm. More detailed characterizations of each region can be found in Supplementary Fig. 1d and Supplementary Note 2. Figure 1c schematically shows that the target material is selectively deposited on the $MoSe_2$ region but not on the $MoS_2$ region after the conventional ALD process. We used the AS-ALD target materials, such as $Al_2O_3$, $HfO_2$, Ru, $Sb_2Se_3$, and Te (Fig. 1c, blue), as shown in Supplementary Fig. 2. Figure 1d is a representative SEM image taken after $Al_2O_3$ deposition by thermal ALD on the $MoS_2$-$MoSe_2$ lateral superlattice. The white areas are selectively deposited $Al_2O_3$ on $MoSe_2$, and the black areas are $MoS_2$ with nothing on them. Below, we describe the line pattern control (Fig. 2), initial deposition site of SAS-ALD (Fig. 3), and mechanism of SAS-ALD (Fig. 4).

### Characterization and controllability of SAS-ALD

Figure 2a shows the atomic force microscopy (AFM) images of the structure in which $Al_2O_3$ is selectively deposited on the $MoS_2$-$MoSe_2$ lateral superlattice grown on a Si/$SiO_2$ (300 nm) substrate. We use TMA and $H_2O$ as ALD precursors, and the deposition temperature is 170 °C (see "Methods" for the details). Through three-dimensional tilted and top view images (Fig. 2a; top, middle), we confirmed that $Al_2O_3$ is selectively deposited with a 70 nm width and a 120 nm pitch size. The height profile (Fig. 2a; bottom) shows that the $Al_2O_3$ thickness is about 10 nm. The structure can also be identified with a tilted SEM image (Supplementary Fig. 3a). Figure 2b shows a cross-section high-angle-annular-dark-field scanning transmission electron microscopy (HAADF-STEM) image of the structure, which is similar to that shown in Fig. 2a but with a narrower width (~ 25 nm). $Al_2O_3$ is periodically and selectively deposited on the monolayer $MoS_2$-$MoSe_2$ lateral superlattice. A cross-section energy dispersive X-ray spectroscopy (EDS)

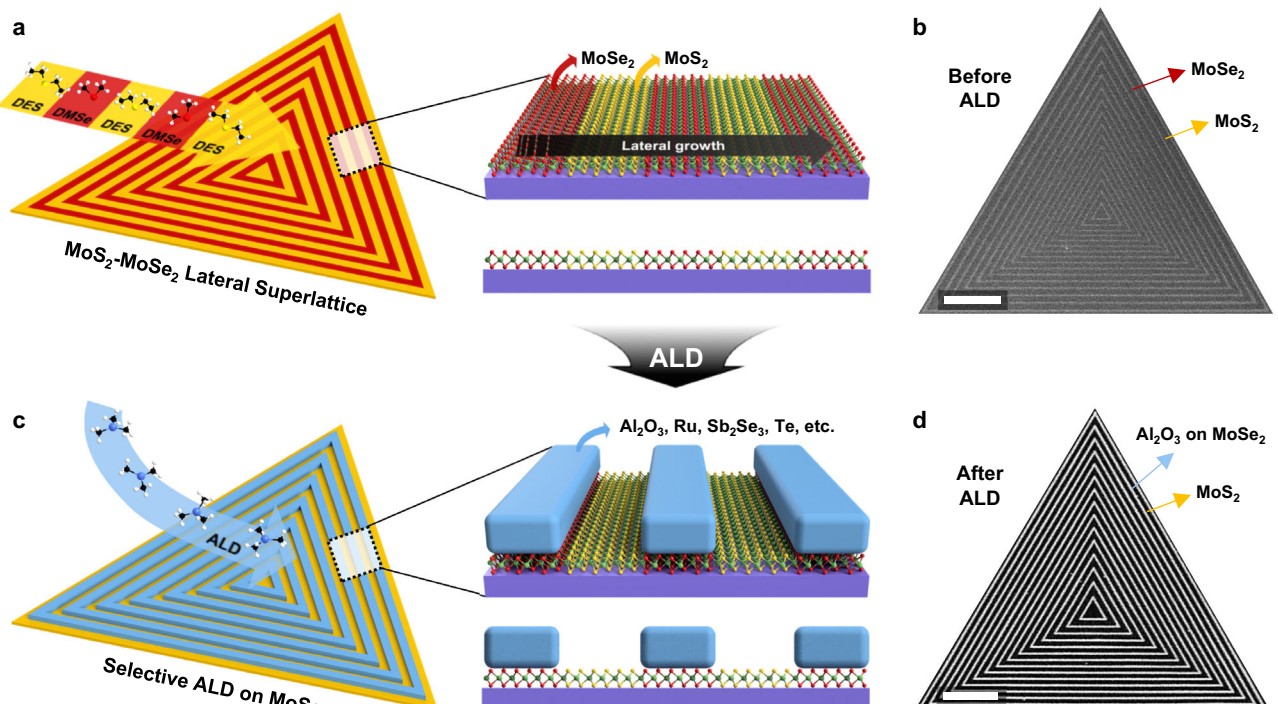

**Fig. 1 | Superlattice-based area-selective atomic layer deposition (SAS-ALD) process.** Schematic (**a**) and scanning electron microscopy (SEM) image (**b**) of the monolayer $MoS_2$-$MoSe_2$ lateral superlattice by chemical vapor deposition (CVD) process with diethyl sulfide (DES) and dimethyl selenide (DMSe). **c** Schematic for SAS-ALD on $MoSe_2$ region of lateral superlattice. The deposited materials can be $Al_2O_3$, $HfO_2$, Ru, $Sb_2Se_3$, and Te. **d** SEM image after $Al_2O_3$ SAS-ALD on the $MoS_2$-$MoSe_2$ lateral superlattice. Scale bar for (**b**) and (**d**), 1 μm.

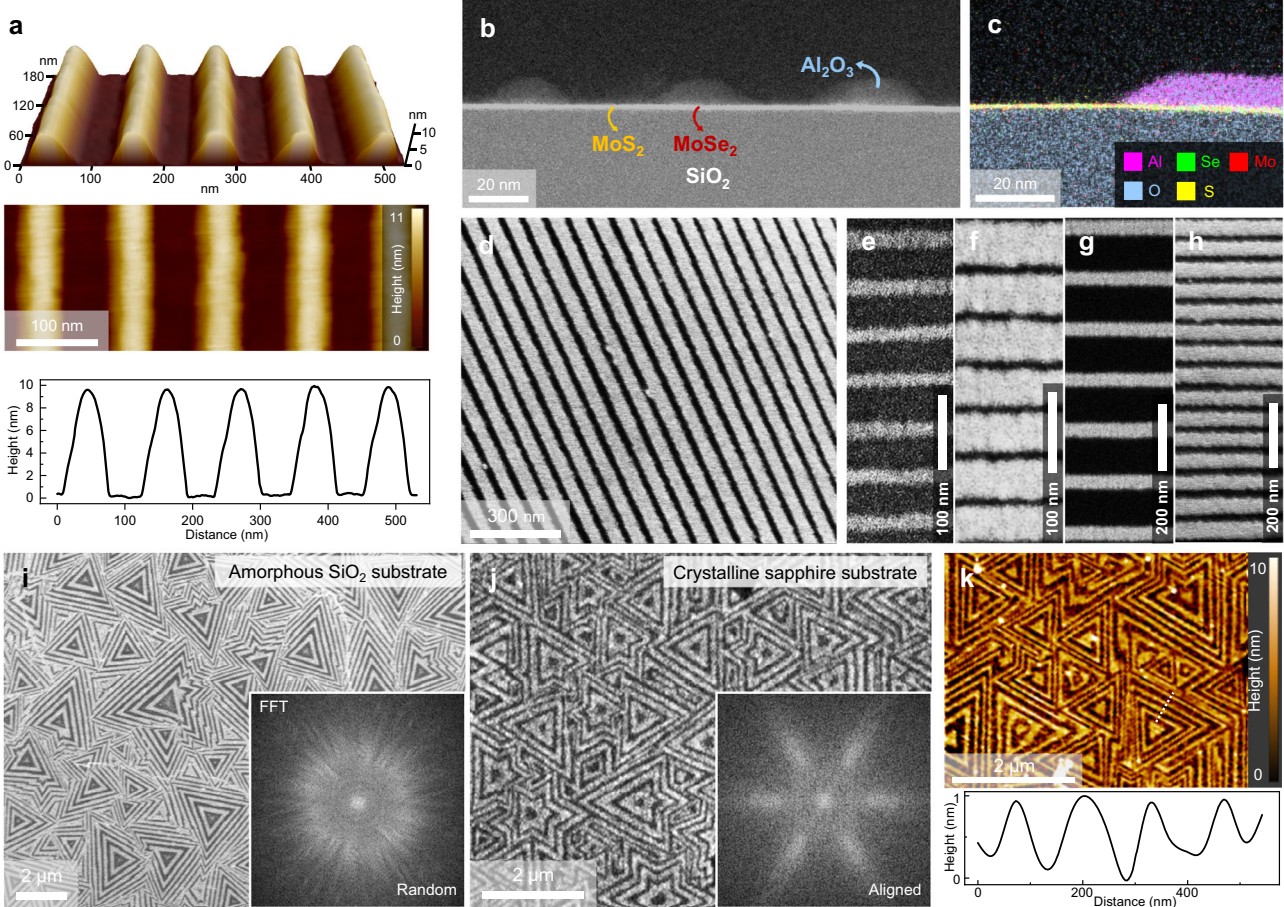

**Fig. 2 | Characterization of Al$_2$O$_3$ SAS-ALD on MoS$_2$-MoSe$_2$ lateral superlattice.** **a** Atomic force microscopy (AFM) images of the structure after Al$_2$O$_3$ ALD on the monolayer lateral superlattice, which are displayed in a 3D tilted view (top), top view (middle), and height profile (bottom). Cross-section high-angle-annular-dark-field scanning transmission electron microscopy (HAADF-STEM) (**b**) and energy dispersive X-ray spectroscopy (EDS) mapping (**c**) images of the SAS-ALD structure, which show periodic and selective deposition of Al$_2$O$_3$. SEM images of uniform line pattern (**d**) and patterns with various width-space scales (**e-h**). SEM images of Al$_2$O$_3$ SAS-ALD on amorphous Si/SiO$_2$ (**i**) and c-plane sapphire (**j**) substrate. The inset images are the Fast Fourier Transform (FFT) data. **k** AFM image of SAS-ALD structure with epitaxial lateral superlattice. The below graph shows the height profile of white dotted line in the AFM image.

mapping image in Fig. 2c shows that aluminum atoms are detected on a selenium rich-region (right, MoSe$_2$) but not on a sulfur-rich region (left, MoS$_2$), which indicates that Al$_2$O$_3$ is selectively deposited on the MoSe$_2$ region only. Additionally, the interface between Al$_2$O$_3$ and MoSe$_2$ is clean, and the aluminum oxide exhibits valid bandgap values (see Supplementary Note 8).

Because of lateral growth, a crystallographic phenomenon exhibited by 2D materials, our SAS-ALD technique allows nanoscale control over the width of the patterning template (lateral superlattice) by adjusting the duration time of the gas-phase chalcogen precursors. Figure 2d–h shows line pattern controllability by adjusting the width of the superlattice. It is possible to form a uniform line pattern of Al$_2$O$_3$ spanning a few micrometers (Fig. 2d) and control the width and pitch size of the patterns (Fig. 2e–h). Figures 2e and 2f have the same pitch size (~57 nm) but different width sizes of Al$_2$O$_3$, 15 nm in Fig. 2e and 48 nm in Fig. 2f. Figures 2g and 2h have the same width size of Al$_2$O$_3$ (~ 45 nm) but different pitch sizes, 155 nm in Fig. 2g and 70 nm in Fig. 2h. In addition, Supplementary Fig. 3d shows patterns with the same pitch size (~96 nm) but with reversed width/space ratios (1:2 and 2:1). Supplementary Fig. 3e describes various width/space ratios ranging from 0.25 to 13. The minimum pitch size of the Al$_2$O$_3$ pattern is 19.7 nm (width ~ 16 nm), as shown in Supplementary Fig. 3f. Furthermore, in SAS-ALD, selectivity is maintained up to 15 nm of Al$_2$O$_3$ thickness (Supplementary Fig. 3b). From the above data, SAS-ALD can control various line patterns of target material with uniformity and

high selectivity. The line patterns in SAS-ALD can be controlled diversely within a single flake (Fig. 2a–h), but the orientation between flakes is not aligned. Typically, as shown in Fig. 2i, lateral superlattices grown on a Si/SiO$_2$ substrate have different directions, resulting in randomly oriented Al$_2$O$_3$ line patterns, as confirmed in the Fast Fourier Transform (FFT) image. In contrast, when utilizing a c-plane sapphire substrate, which facilitates the epitaxial growth of TMDs, the orientations of lateral superlattice flakes can be aligned, allowing for directional alignment of Al$_2$O$_3$ line patterns, as seen in Fig. 2j. Figure 2k presents AFM analysis for Fig. 2j, showing the selective deposition of 1 nm Al$_2$O$_3$ on the epitaxial lateral superlattice film. Lastly, the Al$_2$O$_3$ SAS-ALD is possible not only for the MoS$_2$-MoSe$_2$ lateral superlattice mentioned above but also for a WS$_2$-WSe$_2$ lateral superlattice. The WS$_2$-WSe$_2$ superlattice was confirmed through Raman spectroscopy and SEM images (Supplementary Fig. 19a–c), and the Al$_2$O$_3$ was selectively deposited on the WSe$_2$ regions (Supplementary Fig. 19d).

Throughout Figs. 1, 2, we provide experimental evidence confirming the selective ALD occurring specifically on the MoSe$_2$ region within the MoS$_2$-MoSe$_2$ lateral superlattice. This phenomenon of selectivity occurring on the basal planes of a 2D material with a non-dangling bond is totally different from the CAS-ALD, in which selectivity arises from the chemical reaction of the surface. In order to examine the mechanism of SAS-ALD, we analyzed the initial nucleation sites where deposition primarily occurs (Fig. 3) and performed relevant simulations (Fig. 4).

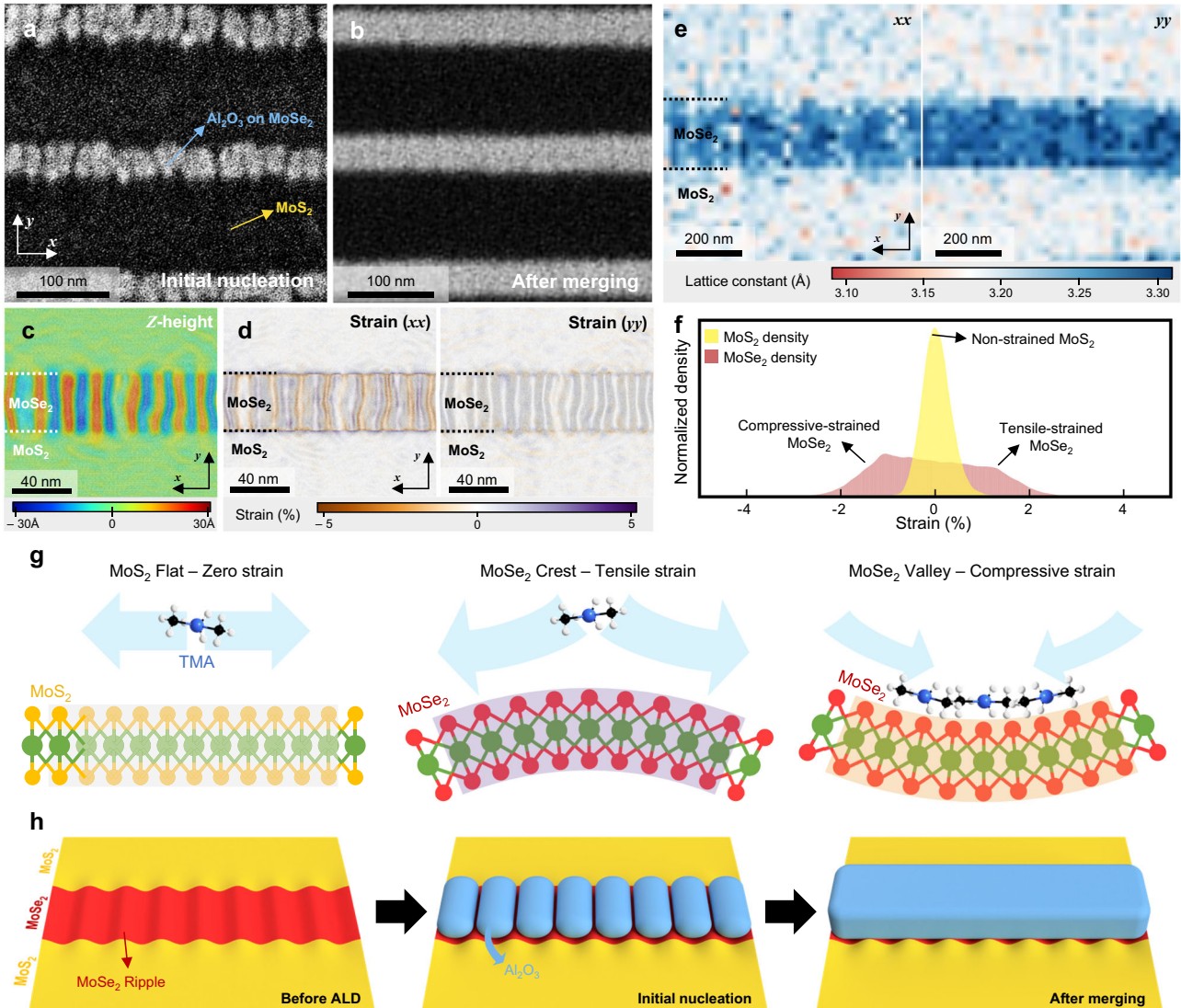

**Fig. 3 | Periodic deposition of Al₂O₃ on MoSe₂ region in SAS-ALD.** SEM images of initial nucleation step with 15 ALD cycles (**a**), and after merging step with 60 ALD cycles (**b**) on MoS₂-MoSe₂ lateral superlattice. Periodic Al₂O₃ deposition (bright region) can be seen in (**a**). **c, d** Molecular dynamics (MD) simulation mapping images of the MoS₂-MoSe₂ lateral superlattice. The dotted lines are boundaries between MoS₂ and MoSe₂ regions. Z-height (**c**) and strain (**d**) periodically change in MoSe₂ region. **e** Lattice constant mapping data of the lateral superlattice by electron microscope pixel array detector (EMPAD) 4D STEM. **f** The *xx* and *yy* averaged strain density plot of the lateral superlattice cell in MD simulation. Yellow (red) region is MoS₂ (MoSe₂) density. **g** Schematics of three surfaces in the lateral superlattice; non-strained MoS₂, tensile-strained MoSe₂ crest, and compressive-strained MoSe₂ valley. The molecules on the surfaces are trimethyl aluminum (TMA). **h** Schematics for process of SAS-ALD. Initial nucleation of Al₂O₃ periodically occurs in MoSe₂ region, and the Al₂O₃ islands are merged.

## Initial nucleation sites in SAS-ALD

Figure 3h schematically represents a SAS-ALD process comprised of before ALD, initial nucleation, and after merging steps. Before ALD, periodic ripple structure forms within the MoSe₂ region of the lateral superlattice, as shown in Supplementary Fig. 1e. Figure 3a shows a SEM image of a lateral superlattice with 15 Al₂O₃ ALD cycles, in which the light gray region shows Al₂O₃ deposited on MoSe₂ and the dark region shows MoS₂ without any deposition. Initial nucleation of Al₂O₃ forms islands, with each island nucleating periodically in the *x*-direction (under 20 nm in periodicity). When 60 ALD cycles are performed, the Al₂O₃ islands fully merge, forming a continuous line, as shown in Fig. 3b. Interestingly, Al₂O₃ not only selectively deposits on the MoSe₂ region, but also exhibits preferential and periodic deposition at specific sites within the MoSe₂. Supplementary Fig. 4a–d show the SAS-ALD process in a lateral superlattice with a relatively wider MoSe₂ width. In this case, as shown in Supplementary Fig. 4a, large buckle structures of a few nanometers periodically form within the MoSe₂

region. In addition, in Supplementary Fig. 4b, periodic Al₂O₃ deposition similar to Fig. 3a can be observed. A detailed explanation of the periodic deposition of Al₂O₃ is provided in Supplementary Note 3.

The periodic deposition observed in Fig. 3a and Supplementary Fig. 4b closely resemble the periodicity of ripples and buckles in the lateral superlattice. To understand the influence of these structures, we conducted Molecular Dynamics (MD) simulations and 4D STEM measurements. Figures 3c and 3d show the *z*-directional height and strain mapping of the superlattice in MD simulation. In the *z*-height mapping (Fig. 3c), while the MoS₂ region remains relatively flat, the MoSe₂ region is found to form repetitive ripples to relieve the compressive strain that results from coherent bonding with MoS₂, which has a smaller lattice constant[21]. This ripple structure can be measured by AFM in Supplementary Fig. 1e. Importantly, in the strain mapping (Fig. 3d), a periodic compressive-tensile lattice strain exists in the MoSe₂ region, aligning with the ripple structure. The strain in Supplementary Fig. 5a shows the average atomic strain

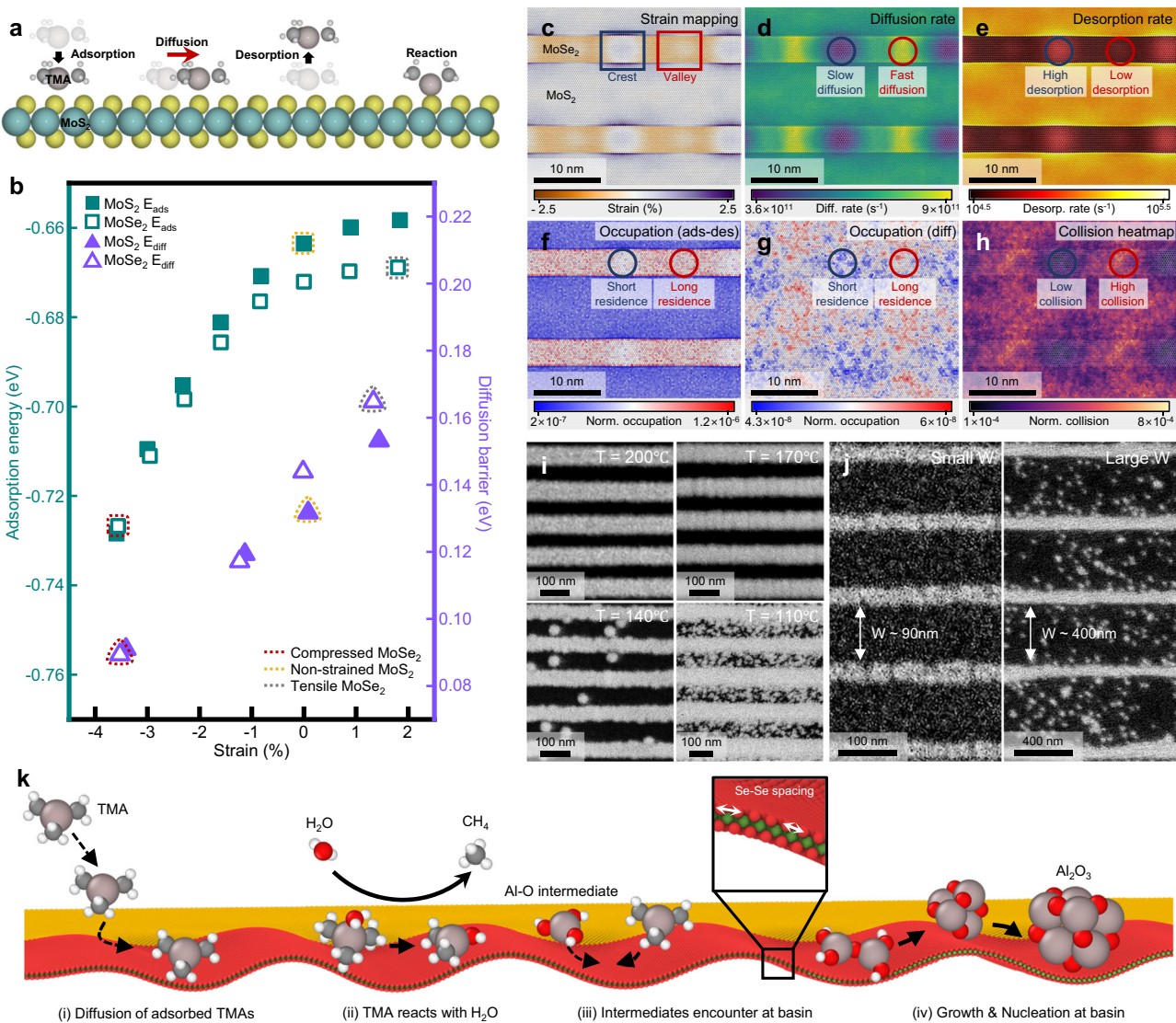

**Fig. 4 | SAS-ALD mechanism. a** Schematics of TMA behaviors on 2D surface; adsorption, diffusion, desorption, and reaction. **b** Adsorption energy (green) and diffusion barrier (purple) plot of TMA molecule on $MoS_2$ (filled dot) and $MoSe_2$ (empty dot) surface with varying strain. Red, yellow, and gray dotted lines mean compressive-strained $MoSe_2$, non-strained $MoS_2$, and tensile-strained $MoSe_2$, respectively. Visualizations of strain (**c**), diffusion rate (**d**), desorption rate (**e**),

occupation with adsorption-desorption mode (**f**), occupation with diffusion mode (**g**), and collision (**h**) of TMA on $MoS_2$-$MoSe_2$ lateral superlattice cell at T = 443 K obtained by kinetic Monte Carlo (kMC) simulation. The red (blue) square and circle regions are $MoSe_2$ valley (crest) regions. SEM images of $Al_2O_3$ SAS-ALD with changing ALD temperature (**i**) and width scale of $MoS_2$ region (**j**). **k** Schematics of diffusion effect in SAS-ALD mechanism.

tensor on S and Se atoms. The Se atoms near the crest of ripples are under tensile strain, whereas Se atoms near the valley are under compressive strain. The density plot for population of the *xx* and *yy* average strain is described in Fig. 3f. From the density plot, we find that most strains on the $MoS_2$ (yellow) are released, whereas the $MoSe_2$ (red) gets both compressive and tensile strains. More information about strain mapping by MD simulation is provided in Supplementary Note 4. In addition, the STEM data of lattice constant mapping in Fig. 3e provide a rational explanation for the formation of the repetitive strain. In the *x*-directional lattice constant mapping of Fig. 3e, white vertical line regions exist within $MoSe_2$, indicating the presence of periodic compressive strain sites (reduction in the lattice constant) in $MoSe_2$. Based on the MD simulations and STEM measurements, the $MoS_2$-$MoSe_2$ lateral superlattice can be divided into three regions as shown in Fig. 3g: non-strained $MoS_2$, tensile-strained $MoSe_2$ crest, and compressive-strained $MoSe_2$ valley. Furthermore, the periodic initial deposition on $MoSe_2$ in SAS-ALD is closely related to this repetitive strain in the $MoSe_2$ region.

## Mechanism of SAS-ALD

To begin with, we examine the possible interactions and behaviors of ALD precursors in the lateral superlattice, prior to investigating the relation between the SAS-ALD process and periodic strain in the $MoSe_2$ region. Figure 4a schematically illustrates four representative processes (adsorption, diffusion, desorption, and dissociative reaction) of the ALD precursor (TMA) on the TMD. In CAS-ALD, a specific reaction can be considered as the key to selectivity, where the reaction rate between the ALD precursors and substrate is faster in deposition region than barrier region. However, our lateral superlattice, unlike conventional ALD substrates, lacks dangling bonds on the surface, rendering it chemically inert to direct reactions. Consequently, direct reactions are not expected to be the primary factor influencing selectivity. Indeed, we observe that the direct dissociative reaction of TMA to the TMD surface is highly endothermic on both areas of the superlattice surface (1.59 eV in $MoS_2$, 2.03 eV in $MoSe_2$, see Supplementary Table 1). In the reaction of $H_2O$, the energies are more unstable (5.09 eV in $MoS_2$, 5.22 eV in $MoSe_2$). Additionally, the

substantially higher endothermic reaction energy on $MoSe_2$ than on $MoS_2$ presents a contradiction to our SAS-ALD results.

Given these points, the influence of defects in the superlattice must also be examined. Defects in TMDs can disrupt the surface inactivity of ALD precursors, therefore requiring consideration of this factor as well. In our case, the CVD-grown TMDs inevitably contain defect sites, primarily chalcogen vacancies, as shown in the HAADF-STEM images in Supplementary Fig. 1d. However, we observe that the density of defects does not exhibit periodicity in the $MoSe_2$ region, suggesting that defects are not the primary factors controlling selective deposition (Supplementary Note 2). Nevertheless, as defect sites can influence nucleation behavior, we included chalcogen vacancy sites in our investigation. Supplementary Table 1 provides a list of possible processes for ALD on the superlattice, along with the lowest energy configurations obtained for each process. We found that TMAs adsorb to the chalcogen top sites with adsorption energies of −0.67 and −0.66 eV on $MoSe_2$ and $MoS_2$ surfaces, respectively. On chalcogen vacancies, the binding is significantly weakened, measuring −0.48 eV for both TMD surfaces. This indicates that vacancies do not exhibit a stronger affinity for TMA adsorption. The adsorption of $H_2O$ is relatively weak compared to TMA on the chalcogen top sites (−0.15 eV on $MoS_2$ and $MoSe_2$) and the chalcogen defect sites (−0.25 eV on $V_S$ and −0.30 on $V_{Se}$). However, we found that $H_2O$ adsorption on pre-adsorbed TMA exhibits stronger adsorption energies (−0.43 eV on $MoSe_2$ and $MoS_2$, see Supplementary Fig. 11). Additionally, the consecutive reaction of the TMA-$H_2O$ complex shows a small activation energy (~ 0.34 eV) and a highly endothermic reaction energy (~ −0.94 eV), implying rapid reaction once such a complex form. This means that the $H_2O$ will follow the adsorption selectivity of TMA, not determining selective deposition itself. In terms of dissociative reactions between ALD precursors and chalcogen vacancy sites, the reaction energy of TMA is 1.09 eV on $V_S$ and 1.38 eV on $V_{Se}$ (Supplementary Table 1). In the case of $H_2O$, the reaction energy is 1.22 eV on $V_S$ and 1.29 eV on $V_{Se}$. All of the dissociative reactions exhibit a very high energy level (1 ~ 2 eV), making them hard to occur easily. Importantly, even the dissociative reaction of ALD precursors at chalcogen vacancy sites is more favorable in $MoS_2$ region than $MoSe_2$ region, which contradicts our SAS-ALD results. Therefore, considering the distribution of defects and the adsorption and reaction energies of the ALD precursors at the chalcogen vacancy sites, we can rule out the possibility that defects induce selective deposition in SAS-ALD. More detailed explanation of defect and reaction effects is provided in Supplementary Note 5.

In contrast to CAS-ALD, the reactions of ALD precursors on the substrate or any defect sites are not the origin of the selectivity in SAS-ALD. Therefore, we need to analyze other behaviors of the precursors (Fig. 4a), and the first one is adsorption. In Fig. 3, we confirmed that the SAS-ALD is significantly associated with the periodic strain in the $MoSe_2$ region. We thus analyze the adsorption energy of TMA with respect to TMD's lattice straining through DFT calculations. The green square dots in Fig. 4b represent the adsorption energy of TMA molecules as a function of strain on $MoS_2$ (filled square) and $MoSe_2$ (empty square). In both $MoS_2$ and $MoSe_2$, the adsorption energy of TMA stabilizes with increasing compressive strain. Supplementary Fig. 9 illustrates the TMA adsorption energy on TMD surface with respect to the direction of strain, confirming the stability of adsorption with compressive strain. When considering the three areas in Fig. 3g ($MoS_2$ flat, $MoSe_2$ crest, and $MoSe_2$ valley), the adsorption energy of TMA in the compressed $MoSe_2$ valley region (red dotted square in Fig. 4b) is more than 60 meV stable than the other two areas. In addition, in the $MoSe_2$ valley, the adsorption energy of TMA can be further stabilized with curvature. Supplementary Fig. 10 shows the variation of TMA adsorption energy in $MoSe_2$ planar and concave cell, confirming a decrease in adsorption energy in the concave valley. In other words, it can be observed that TMAs bind more strongly to the sites in the

compressively strained $MoSe_2$ valley than in other regions of the lateral superlattice. This phenomenon is attributed to the fact that the interaction of TMA with the TMD is dominantly the van der Waals interaction, with denser/concave surfaces producing stronger interaction. As such, the trends in adsorption energies holds for other AS-ALD materials, with the adsorption of the precursors being most stable in the compressed $MoSe_2$ valley region (Supplementary Figs. 12 and 13. See more details in Supplementary Note 6).

Through the above data, it was found that the TMA adsorption energy is lowest in the compressed $MoSe_2$ valley region. However, relying solely on adsorption energy to explain the high selectivity of SAS-ALD may not provide a complete picture. As shown in Supplementary Fig. 15, when ALD is performed on a single $MoS_2$ or $MoSe_2$ surface, the coverage of $Al_2O_3$ is higher on $MoSe_2$ than $MoS_2$ due to the difference in TMA adsorption energy. However, $Al_2O_3$ clearly deposits on the single $MoS_2$ flake. To explain the high selectivity of SAS-ALD, additional analysis of TMA diffusion was conducted. The purple triangle dot in Fig. 4b shows the TMA diffusion barrier on $MoS_2$ (filled triangle) and $MoSe_2$ (empty triangle) surface as obtained by Nudged Elastic Band (NEB) calculations. Similar to adsorption energy, the diffusion barrier of TMA tends to decrease with compressive strain on the TMD surface, and among the three regions in Fig. 3g, the compressed $MoSe_2$ valley shows the lowest diffusion barrier of TMA ( ~ 0.09 eV, red dotted triangle in Fig. 4b). In other words, TMA is expected to have the most stable adsorption and also the lowest diffusion barrier in the $MoSe_2$ valley area, making the statistical presence of precursors higher in this region.

While it is apparent that DFT-calculated energies are in good agreement with experimental observations, one may point out that the changes in adsorption and diffusion energies are relatively small. However, even minor energetic differences can have profound implications when examining kinetics on larger spatial and temporal scales. To quantify the extent of these differences, we implemented a straightforward kinetic Monte Carlo (kMC) simulation where TMAs adsorb/desorb, and diffuse on a small lateral superlattice generated via molecular dynamics. Figure 4c shows the $xx$, $yy$, and $zz$ strain tensor average mapping of our simulation cell, showing periodic strain distributions in the $MoSe_2$ regions. The red square region represents the $MoSe_2$ valley under compressive strain (orange). The cell configuration and $z$-coordinate ($z$-height mapping) for the lateral superlattice can be seen in Supplementary Fig. 16a and b. Employing linear regression, we derived estimates for the diffusion (Fig. 4d) and desorption (Fig. 4e) rates of TMA across all surface sites on the lateral superlattice cell at $T = 443$ K. Surprisingly, as observed in Fig. 4d, the diffusion rate of TMA is highest in the $MoSe_2$ valley (red circle), approximately three times faster than in the $MoSe_2$ crest (blue circle). In Fig. 4e, the desorption rate of TMA is the slowest in the $MoSe_2$ valley (red circle), with the $MoS_2$ region exhibiting desorption roughly ten times faster than the $MoSe_2$ region. Supplementary Fig. 16c−e provide diffusion and desorption rates, as well as rate constants at $T = 383, 413, 443$, and 474 K. It can be observed that as the temperature increases, both diffusion and desorption rates generally become faster. Nevertheless, at all temperatures, it is evident that TMA diffusion is fastest and desorption is slowest in the $MoSe_2$ valley area, suggesting a strong direct indication that TMA is more abundant and likely to collide more frequently in the $MoSe_2$ valley region. Using our estimated rate constants, we conducted kMC simulations of a TMA pulse. Figure 4f and Fig. 4g present occupation heatmaps at $T = 443$ K, the normalized residence time of TMA molecules at specific sites by total simulation time. Figure 4f represents the adsorption-desorption mode, while Fig. 4g is the diffusion-only mode for TMA. Importantly, both of these occupation mappings confirm the long residence of TMA in the $MoSe_2$ valley area (red circle). In addition to TMA occupation, we also examined the frequency of collisions between TMA molecules at $T = 443$ K (Fig. 4h). The collision heatmap visualizes the number of TMA collisions

occurring at each site during simulation, normalized by the total collision number. Since the diffusion of TMA molecules on the TMD surface is very rapid, the collision frequency of TMA can also serve as a good estimate of nucleation in SAS-ALD, given the structural similarities between the TMAs and the reactive single $Al(OH)_n(CH_3)_{3-n}$ intermediates. As evident in Fig. 4h, the $MoSe_2$ valley region experiences the highest frequency of TMA collisions. In summary, the $MoSe_2$ valley region is predicted to have the highest TMA presence in terms of adsorption, diffusion, and desorption. Based on this calculation, the kMC simulations show the longest TMA residence and highest collision rate on the compressed $MoSe_2$ valley area.

The control experiments also provide evidence of the significant roles played by TMA's adsorption and diffusion in the SAS-ALD process. Figure 4i shows SEM images depicting the trends in $Al_2O_3$ AS-ALD with varying ALD temperatures. When the temperature is relatively high (200 and 170 °C), undesired deposition does not occur on the $MoS_2$ region (black area). However, as the temperature decreases (140 and 110 °C), $Al_2O_3$ deposition occurs on the $MoS_2$ region as well. In other words, when the temperature is low, the selectivity in SAS-ALD diminishes. This phenomenon is expected to occur because as the temperature decreases, the distance that TMAs can move is reduced, resulting in many TMA molecules being present on the $MoS_2$ region as well. In particular, as the temperature decreases, the coverage of TMA on the whole superlattice cell increases (Supplementary Fig. 17) and the collision rate of TMA on the cell also increases (Supplementary Fig. 18). This suggests that while TMA diffusion is quite fast, at lower temperatures, a significant number of TMA molecules exist on the $MoS_2$ region and form nucleation before desorbing or reaching the $MoSe_2$ valley area, where more nucleates will be present. As a result, at lower ALD temperatures, the higher presence of TMA on the $MoS_2$ region, along with an increased frequency of collisions, leads to many undesired depositions. Such an explanation is also backed by the fact that the $MoS_2$ width also shows strong correlation to undesired deposition. The $MoS_2$ width is considered the distance over which TMA should diffuse for AS-ALD, and as the width of $MoS_2$ increases, more TMAs fail to arrive at the $MoSe_2$ valley, leading to increased undesired deposition on $MoS_2$ region, as shown in Fig. 4j.

Our findings confirm that the SAS-ALD mechanism can be represented by a complex interplay of various interactions, such as adsorption, diffusion, and desorption of ALD precursors on 2D TMD surfaces, with the compressed $MoSe_2$ valley region identified as the primary site for deposition, as shown in Fig. 3h. The selectivity inherent in SAS-ALD cannot be attributed to reactions between TMA and the 2D TMD surfaces or to the presence of chalcogen vacancy sites (Supplementary Note 5). Instead, the selectivity is originated by the energetically most stable adsorption (Fig. 4b) and the slowest desorption rates (Fig. 4e) in the $MoSe_2$ valley region, leading to the highest concentration of TMA. With respect to diffusion, the $MoSe_2$ valley presents the lowest surface diffusion barrier for TMA, as shown in Fig. 4b, corresponding with the fastest diffusion rates in this region, as depicted in Fig. 4d. The kMC simulations support this, suggesting that TMA's residence time is longest in the $MoSe_2$ valley, as detailed in Figs. 4f and 4g, and collisions are most frequent in this area, as indicated in Fig. 4h. This evidence highlights AS-ALD mechanism within SAS-ALD, wherein diffusion plays a crucial role, as represented schematically in Fig. 4k.

## Discussion

In conclusion, we have discovered AS-ALD process on the 2D lateral superlattice surfaces, which is predominantly governed by the adsorption and diffusion of ALD precursors. This distinctive selective deposition mechanism shows promise for application across a variety of materials, including $Al_2O_3$, $HfO_2$, Ru, $Sb_2Se_3$, and Te. Additionally, by employing the lateral growth attributes of the superlattice as a selective deposition template, we have successfully minimized the pattern size to a half-pitch of 10 nm and anticipate the potential to scale down. We also suggest that the combination of 2D superlattices with AS-ALD materials opens the way for fabricating intricate 2D/3D nanoscale electrical devices that were previously challenging to produce. For instance, TMD nanoribbon device can be easily fabricated with SAS-ALD structures (see Supplementary Note 9). Also, when using AS-ALD materials as gate oxide or metal contact for the underlying 2D semiconductors, our 2D/3D structures can be implemented in advanced 2D devices such as short-channel device. To achieve this, it is necessary to confirm and improve the characteristics of AS-ALD materials.

## Methods

### Chemical vapor deposition (CVD) of 2D monolayer lateral superlattice

The monolayer $MoS_2$-$MoSe_2$ lateral superlattice was grown in a home-made horizontal hot-wall quartz tube, with $MoO_3$ powder (99.95%, Alfa Aesar), diethyl sulfide (DES, Sigma Aldrich), and dimethyl selenide (DMSe, Alfa Aesar) as CVD precursors. The mixture powder of $MoO_3$ and KCl (Sigma Aldrich) was located near Si/$SiO_2$ (300 nm) substrate, and chalcogen precursors (DES, DMSe) were injected in gas-phase for the width control of lateral superlattice. The flow rate was controlled by the mass flow controller (MFC). The growth temperature was 650–700 °C, the growth time was 30–60 min, and the pressure is about 1 torr. After reaching the target temperature, chalcogen precursors were injected alternatively using MFC, carried out in four steps. In the $MoS_2$ growth step, Ar/DES (110/10 sccm) was injected with a duration time (Supplementary Fig. 1a, $t_S$) of 10–300 s depending on the target $MoS_2$ width. In the $MoS_2$ purging step, the chalcogen flow was turned off for 2–8 s (Supplementary Fig. 1a, $t_{S,pur}$). In the $MoSe_2$ growth step, Ar/DMSe/$H_2$ (110/10/2 sccm) was injected with a duration time (Supplementary Fig. 1a, $t_{Se}$) of 10–300 s depending on the target $MoSe_2$ width. In the $MoSe_2$ purging step, the chalcogen flow was turned off for 2–8 s (Supplementary Fig. 1a, $t_{Se,pur}$). These four steps make up one cycle, and the number of cycles was set to the total growth time divided by a single cycle time. The width of $MoS_2$ and $MoSe_2$ can be controlled between ten to hundreds nm scale as shown by the examples of Supplementary Fig. 1b, c. For the epitaxial growth of the lateral superlattice, c-plane sapphire substrates were used, and the growth conditions were similar to those mentioned above. The monolayer $WS_2$-$WSe_2$ lateral superlattice was grown by metal organic chemical vapor deposition (MOCVD) with tungsten hexacarbonyl, DES, DMSe, sodium propionate, and oxygen. The growth temperature was 700–730 °C, the growth time was 2 h, and the pressure is under 0.1 torr. The chalcogen precursors (DES and DMSe) were alternatively injected into the chamber, similar to the $MoS_2$-$MoSe_2$ case, with a constant flow of tungsten hexacarbonyl.

### Atomic layer deposition (ALD) condition for superlattice-based AS-ALD

The conventional thermal ALD was used to deposit oxides ($Al_2O_3$, $HfO_2$). $Al_2O_3$ was deposited at 170 °C. Trimethyl aluminum (TMA) was used as an Al source, and $H_2O$ was used as an oxygen source. Pulse times of TMA and $H_2O$ were 0.2 s, and purge times were 10 s via Ar gas flow of 100 sccm. $HfO_2$ was deposited at 170 °C. Tetrakis (dimethylamido) hafnium (IV) was used as a Hf precursor and $H_2O$ was used as an oxygen source. Pulse time of Hf precursor and $H_2O$ was 1.1 s and 0.1 s, respectively, and purge time was 17 s via $N_2$ gas flow of 150 sccm. Ru was grown using $Ru(C_7H_9)(C_7H_7O)$ and $O_2$ as the Ru source and the reactant, respectively[25]. 500 sccm of Ar was used as both Ru precursor carrier gas and purge gas, and $O_2$ flow rate was fixed at 500 sccm. The substrate temperature was maintained at 200 °C. 200 cycles of Ru ALD sequence including the following four steps, Ru injection (70 s), Ru purge (5 s), $O_2$ injection (10 s), and $O_2$ purge (5 s) were conducted. $Sb_2Se_3$ was deposited at the substrate temperature of 70 °C. The precursors of Sb and Se were $Sb(OC_2H_5)_3$ and $[(CH_3)_3Si]_2Se$, respectively,

and the precursors were carried into the process chamber with 50 sccm of Ar gas. 200 sccm of Ar gas was used as purge gas and the substrate temperature was maintained at 70 °C. $Sb_2Se_3$ was formed by repeating 50 cycles of the following steps, Sb injection (2 s), Sb purge (5 s), Se injection (2 s), and Se purge (10 s). Te was deposited at 70 °C by using two precursors; BTMS-Te and $Te(OEt)_4$, held at 40 and 50 °C, respectively. Each precursor, contained in a bubbler-type canister, was delivered with 50 sccm of Ar carrier gas.

## Characterization

The monolayer $MoS_2$-$MoSe_2$ lateral superlattice and AS-ALD structure were measured using field-emission SEM (FE-SEM, Hitachi S-4800). The surface morphology of the superlattices was confirmed by AFM (NX10, Park systems) via contact and non-contact mode. In the contact mode, to prevent sample damage, we applied a very small force (~1 nN) at a low speed (~2 Hz). The TEM analysis was conducted in two different views. For plan view sample preparation, we placed the quantifoil grid upside down such that the holey carbon film surface was attached to the superlattice flakes on a $Si/SiO_2$ (300 nm) substrate. Then, a PDMS piece was placed on top of the grid and pressed down with an appropriate amount of force. Using a pipet, a drop of DI water was cast under the PDMS piece, then the flake was detached from the $Si/SiO_2$ (300 nm) substrate. For cross-section view sample preparation, the cross-sectional TEM samples were prepared using a focused ion beam (FIB, FEI Helios Nano Lab 450HP and Hitachi Triple Beam NX2000). TEM images and EDS spectrum were taken by a FEI Titan3 G2 60–300 at an accelerating voltage of 200 kV for the cross-section view specimen and 80 kV for the plan view specimen. The optical properties of the lateral superlattices were analyzed by confocal Raman spectroscopy (LabRAM ARAMIS) equipped with Ar ion continuous wave laser with 514.5 nm wavelength. The bandgap of aluminum oxide and superlattice film were measured by UV-VIS spectrophotometer (Lambda 1050).

## EMPAD 4D STEM

On top of monolayer $MoS_2$-$MoSe_2$ superlattice, A4 PMMA was spin-coated at 3000 RPM for 1 min. The edges of the substrate were scraped with a razor blade to expose the substrate surface, allowing for faster etching. The substrate was then floated onto a sodium hydroxide solution with a volume ratio of $NaOH:H_2O = 1:10$. After being etched off by the solution, the PMMA film with the superlattice flakes was separated from the substrate. The film was transferred to de-ionized (DI) water and cleaned three to four times to remove any residual solution, then fished out with a piece of copper sheet and air-dried. The desired region was cut off and loaded on the SiN grid window using a sharp tungsten probe mounted in a micromanipulator. To remove PMMA, the SiN grid was annealed in a CVD tube furnace at 150 sccm with a mixture gas of 90 % $N_2$ and 10 % $H_2$. The temperature was raised to 400 °C, maintained at 400 °C for 1 h, and then cooled to room temperature. All 4D-STEM datasets were taken in the Thermo Scientific Titan Themis using an electron microscope pixel array detector (EMPAD)[26]. We used 300 kV and 80 kV for narrow junction and wide junction, respectively. A 0.5 mrad convergence angle was used for both junctions recording every points' diffraction patterns. All the mappings were acquired by utilizing the center of mass calculations. Our previous work contains more experimental details[27].

## MD simulation, DFT calculation, and kMC simulation

All DFT calculations were performed using VASP (Vienna Ab-Initio Software Package) with the projector augmented wave method, using PBE functional for exchange-correlation[28–30]. Plane-wave cutoff energy of 520 eV was used for all calculations. Surface calculations were performed with Monkhorst-Pack sampling of kpoints (3×3×1) with a cell of 30 Å in the z-axis for vacuum spacing, and relaxed until the force acting on each atom was under 0.02 eV/Å[31]. For all molecules and larger cells

for adsorption of bigger ligands, gamma kpoint (1×1×1) was used. The effect of van der Waals force was considered using the D3 method with Becke-Jonson damping[32,33]. For the calculation of reaction and diffusion barriers, the CI-NEB method was used, optimizing the transition state until all forces on the TS are less than 0.02 eV/ Å[34,35]. The MLNEB and DyNEB algorithms were used for efficient calculation[36,37].

DFT calculation of curved TMD surfaces matching that of the experimental length scale is computationally demanding, we simulate subsets of the 2D surface for relaxation. We first generate a periodic $55 \times 1$ $MoSe_2$ corresponding to a wavelength of 182 Å in the x-axis, similar to the average wavelength of ripples found in MD simulations. We then compress the cell with varying x, y lattice compression (0, −2, −4, −6 %) and give sinusoidal amplitudes in the z-direction (0, 9, 12, 15, 18, and 21 Å). From this initial structure, we take a $14 \times 1$ $MoSe_2$ strip and apply vacuum in the x and z-direction, which generates Mo-edge and Se-edge at each x-ends. For realistic relaxation of the atoms at the middle of the TMD surface which retains the enforced lattice strain, at this stage, we fix the Mo atoms at the edges to their initial position and optimize the geometry. From all lattice compression, we take the most stable cases to again take a subset of 9 repeating $MoSe_2$ units at the center ([0,0], [−2,9], [−4,18], [−6,18], where the numbers in each square bracket denote the imposed lattice compression and the given initial amplitude, respectively), and enlarge the cell in the y-axis by 3 to obtain the final 9×3 $MoSe_2$ surfaces. On these surfaces, atoms of two $MoSe_2$ units from both x-ends are fixed to the initial configuration, and the adsorption energy of TMA was calculated. At the edges, two repeating units of $MoSe_2$ were clamped to their original positions before the cell was further relaxed. All vacuum has been set to be larger than 15 Å. K-points were set to be $1 \times 5 \times 1$ and $1 \times 3 \times 1$ for $14 \times 1$ $MoSe_2$ strip and 9×3 $MoSe_2$ strip respectively. Molecular dynamics simulation was carried out using LAMMPS (ver. 17 Feb 2022), using modified Stillinger-Weber potentials developed for 2D TMDs[38,39]. To simulate strain arising from lattice mismatch in the lateral superlattice, we construct monolayer lateral $MoS_2$-$MoSe_2$ structures. Given that no substrate was included in the simulation, the monolayer was able to deform freely, potentially leading to conformations divergent from experimental observations. In order to minimize such deformations while retaining the effects of lattice mismatch, after energy minimization of the cell with the conjugate gradient method, NPT simulations were performed at a low temperature of 4.2 K and 1 bar pressure in x and y direction for 100 ps, as had been suggested to exclude the intervention of thermal deformation and drifting[40,41]. Although the low temperature inevitably excludes the effects of thermal expansion, prior computational studies have indicated that the difference in lattice expansion between $MoS_2$ and $MoSe_2$ is negligible (less than 0.04 %) even at elevated temperatures of 500 K, thereby implying a minimal impact on strain properties[42]. The choice of temperature and timescale resulted in ripples with wavelength in the range of ~170 nm, which is consistent with experimental observations on the 1:1 size matched periodic slab with repetitive stripes of 40 nm and 100 nm $MoSe_2$ and $MoS_2$. Ovito was used for atomic strain analysis and visualization[43]. To obtain strain tensors, planar, unrelaxed cells with cell sizes corresponding to that of pure and energy-minimized $MoSe_2$ and $MoS_2$ were used as reference structures.

To set up the kinetic Monte-Carlo simulation, we recalculate adsorption energies and corresponding structures' vibrational frequencies of TMA adsorption on the p(5×5) TMD surfaces, with $3 \times 3 \times 1$ kpoints, to obtain Gibbs free energies of adsorption and diffusion. The Gibbs free energies for adsorbates were obtained with the hindered translator/hindered rotor model, while the ideal gas model was used for the gas phase TMA as implemented in the Atomic Simulation Environment package (ASE)[43,44]. To estimate the adsorption energy and the hopping barrier at surface sites with arbitrary local strain, we use linear regression on our calculated data of four different strains on $MoSe_2$ and $MoS_2$. As calculation of values at the boundaries in the

superlattice is costly, we use average of the two fitted model ($MoSe_2$ and $MoS_2$).

For the calculation of rates, we follow the methods of outlined in Weckman et al.[45]. The adsorption rates for on arbitrary surface site can be calculated using the kinetic gas theory;

$$k_{ads,i} = P\sigma(T,\theta)A_i / \sqrt{(2\pi m k_b T)} \qquad (1)$$

Where $P, \sigma, A_i, m, k_b$ and $T$ are the pressure, surface area of a site, sticking probability, mass, Boltzmann constant, and temperature, respectively. As we do not consider coverage in our rate model, we set the $\sigma$ to unity.

The desorption rate is inversely proportional to the adsorption rate through Gibbs free energy of adsorption.

$$\Delta G_{ads} = k_{ads,i} / k_{des,i} \qquad (2)$$

Thus, we can obtain the desorption rates for a particular site given an estimate of $\Delta G_{ads}$.

The diffusion rate of TMA can be obtained from the following equation[46].

$$k_{diff,ij} = f_{i,j}^{diff,TST}(T)(k_B T/h) \exp\left(-\Delta E_{0,i,j}^{diff} / k_B T\right) \qquad (3)$$

and

$$f_{i,j}^{diff,TST}(T) = q_{TS,i,j}^{vib} / q_i^{vib} \qquad (4)$$

where $h, \Delta E_{0i,j}^{diff}, q_{TS,i,j}^{vib}$ and $q_i^{vib}$ are the Planck constant and the zero-point energy corrected diffusion barrier from site $i$ to $j$, and partition function at the transition state or site i, respectively. Generally, the pre-factor to the equation can be set to constant for diffusion, as they give negligible differences compared to the exponential term[47], and we set this to $1\times10^{13}$.

Using the obtained reaction rates, we conduct lattice kinetic Monte-Carlo simulation using the Bortz-Kalos-Lebowitz (BKL) algorithm[48]. In execution of the code, we run an adsorption and desorption only simulation for 300000 steps (corresponding to ~25 ms) to equilibrate the surface coverage. Then we run 700000 steps considering only the first-neighbor hop diffusion reactions, where we measure the adsorbate collision frequencies. For more information on the model setup, refer to Supplementary Note 7.

### Device fabrication
The $MoSe_2$ NRFET was fabricated using $Al_2O_3$ SAS-ALD structure on a $Si/SiO_2$ (90 nm) substrate, which is used as a global back gate. The source and the drain electrodes were patterned by electron-beam lithography, and the contact metal of Cr/Au (10/40 nm) was deposited by electron-beam evaporator. Subsequently, the $MoSe_2$ nanoribbon was defined using the reactive ion etching (RIE) process (Ar plasma, 1 min) on the SAS-ALD structure, where the $Al_2O_3$ serves as a hard mask for dry etching. For $Al_2O_3$ encapsulation layer, ALD process was followed to deposit 30 nm $Al_2O_3$ at 150 °C. Electrical characterization was performed using Keithley 4200A-SCS parameter analyzer.

## Data availability
The Source Data underlying the figures of this study are available with the paper. All raw data generated during the current study are available from the corresponding authors upon request. Source data are provided with this paper.

## Code availability
The code and the baseline simulation cell used for our custom lattice kinetic Monte Carlo simulation is available in zenodo data repository[49] under accession code https://doi.org/10.5281/zenodo.10682150.

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

## Acknowledgements

This research was primarily supported by National R&D Program through the National Research Foundation of Korea (NRF) funded by Ministry of Science and ICT (2022M3H4A1A01013228). Additional funding was provided by the National R&D Program through the National Research Foundation of Korea (NRF) funded by Ministry of Science and ICT (2022M3H4A1A01010325, 2020M3D1A1110659, 2020R1C1C1011219, 2021M3H4A6A01041234, 2021M3F3A2A01037738, 2022M3F3A2A01072215, RS-2023-00258309, and 2021M3D1A2O-45614). B.S. and Y.H. acknowledge the support from Welch Foundation (C-2065-20210327).

## Author contributions

K.K and J.P conceived the project. J.P performed lateral superlattice growth by CVD, epitaxial growth, oxide AS-ALD, SEM, and AFM measurements. Y.K, S.-J.K, W. K and W.-B.L analyzed the mechanism by MD, DFT, and kMC simulation. S.Kang made NRFET device and analyzed electrical properties. H.-Y.J and S.O performed the TEM, STEM, EDS measurements, and Y.H, B.S, C.S, and J.Lou performed 4D EMPAD STEM measurements. G.N and J.-Y.K performed HfO2 ALD. J.S, C.K, and N.H performed Te ALD. T.E, T.-M.C, H.P, and S.-H.O performed Sb2Se3, Ru ALD. H.Y and Y.L analyzed AFM measurements. T.-S.K assisted in CVD setup and discussion. H.-J.C assisted in ALD process. J.Lee analyzed optical property by UV-VIS. M.K assisted in AFM measurements. S.Kwon and E.M provided constructive suggestions. K.K and Y.K supervised this project. All authors contributed to the writing of the manuscript.

## Competing interests

The authors declare no competing interests.

## Additional information

**Peer review information** : *Nature Communications* thanks Rong Chen and the other, anonymous, reviewer for their contribution to the peer review of this work. A peer review file is available.

[1]Department of Materials Science and Engineering, Korea Advanced Institute of Science and Technology (KAIST), Daejeon, Republic of Korea. [2]School of Chemical and Biological Engineering and Institute of Chemical Processes, Seoul National University (SNU), Seoul, Republic of Korea. [3]Graduate School of Semiconductor Materials and Devices Engineering, Ulsan National Institute of Science and Technology (UNIST), Ulsan, Republic of Korea. [4]Department of Materials Science and NanoEngineering, Rice University, Houston, TX, USA. [5]Center for Neuromorphic Engineering, Korea Institute Science and Technology (KIST), Seoul, Republic of Korea. [6]Department of Materials Science and Engineering, Ulsan National Institute of Science and Technology (UNIST), Ulsan, Republic of Korea. [7]Division of Advanced Materials, Korea Research Institute of Chemical Technology (KRICT), Daejeon, Republic of Korea. [8]Graduate School of Semiconductor Technology, Korea Advanced Institute of Science and Technology (KAIST), Daejeon, Republic of Korea. [9]Department of Electronic and Electrical Engineering, Ewha Womans University, Seoul, Republic of Korea. [10]Department of Physics, Korea Advanced Institute of Science and Technology (KAIST), Daejeon, Republic of Korea. [11]Department of Materials Science and Engineering, Korea University, Seoul, Republic of Korea. [12]These authors contributed equally: Jeongwon Park, Seung Jae Kwak, Sumin Kang. ✉e-mail: cjyjee@korea.ac.kr; kibumkang@kaist.ac.kr

