## [Peer Review File · Nature Communications]

Area-selective atomic layer deposition on 2D monolayer lateral superlatticesEditorial Note: This manuscript has been previously reviewed at another journal that is not operating a transparent peer review scheme. This document only contains reviewer comments and rebuttal letters for versions considered at *Nature Communications*.

REVIEWER COMMENTS

Reviewer #1 (Remarks to the Author):

The authors have addressed my questions and questions from other reviewers in my view. Publication is recommended.

Reviewer #3 (Remarks to the Author):

In this version of the manuscript, the authors have addressed some of the concerns raised by reviewers previously. Although the electrical properties and scalability that are important to practical devices are not solved, this work is important as it demonstrates a new selective ALD phenomenon. Furthermore, the authors have focused the article on the mechanism of the induced area-selective ALD on 2D materials. The origin of selectivity has been clarified. Therefore, I think this paper is deserved to be published on Nat. Commun. After address the following points.

(1) The authors have included further explanations in the MD simulation part, which significantly improves the manuscript. Yet the thermodynamics effect still needs to be addressed. Significant flexural deformation or drifting can be anticipated if the substrate is NOT CONSIDERED in the model. Please see Al-Mulla, Talal, Zhao Qin, and Markus J. Buehler. Journal of Physics: Condensed Matter 27.34 (2015): 345401. The adhesion energy between the substrate and TMD is a major factor in the deformation. In addition, thermal expansion at a high temperature cannot be represented by the case at 4.2 K. Therefore, I recommend that the authors include the substrate in the model to ensure the reliability of MD results, as well as a schematic of the MD model.

(2) The authors perform significant amount of calculations on the adsorption energy of the stress-induced substrates and vacancies. However, the slight difference of the adsorption energy and energy difference cannot interpret the obvious selectivity. The KMC simulation which includes the adsorption, diffusion and desorption of the ALD precursors shows that the TMA has the longest residence and the most active collisions in the MoSe₂ valley region. The authors attribute the selective deposition to combination the adsorption, diffusion and desorption of the ALD precursors. However, the residence time and collisions are indirectly correlated to the selectivity. The coverage of Al species should be provided by the KMC, which can quantitatively represent the selectivity between the two regions.

(3) Although the authors claim that they focus on uncovering the SAS-ALD mechanism, it is still concerned that the interfaces between the ALD film and substrate need to be described. It is an ongoing question for an ideal interface between ALD film and 2D materials. The interface property would significantly alter the electrical properties of the dielectric oxide films. Therefore, the leakage current, breakdown voltage and EOT needed to be explored to support this work for potential application. It is also notice that the authors have provided contact measurement for previous reviewer's concern, and

the results show that the contact has poor electrical properties. It would be beneficial for the authors to provide possible discussions and solutions to improve the electrical properties via interface engineering.

(4) It is also noticed that another reviewer has raised concerns on the defects and interface region induced selective deposition. The authors have added reasonable discussions about the influence of different defects to selective growth. The grain boundaries region between two different 2D materials are important areas to impact the selective growth. Although the authors try to decouple the influences of the grain boundary, they shall also discuss clearly the initial nucleation stage on the interface areas and on MoSe₂. In addition, it is suggested to demonstrate the ALD nucleation stage on MoS₂-MoSe₂ substrates with larger MoSe₂ width. All these could help to decouple the grain boundaries region effect as raised up by the previous reviewer.

1. Point-by-Point Responses to Reviewer #3 Comments

(1) The authors have included further explanations in the MD simulation part, which significantly improves the manuscript. Yet the thermodynamics effect still needs to be addressed. Significant flexural deformation or drifting can be anticipated if the substrate is NOT CONSIDERED in the model. Please see Al-Mulla, Talal, Zhao Qin, and Markus J. Buehler. *Journal of Physics: Condensed Matter* 27.34 (2015): 345401. The adhesion energy between the substrate and TMD is a major factor in the deformation. In addition, thermal expansion at a high temperature cannot be represented by the case at 4.2 K. Therefore, I recommend that the authors include the substrate in the model to ensure the reliability of MD results, as well as a schematic of the MD model.

[Author reply]

Thermal Expansion:

We are grateful to the reviewer for suggesting the consideration of thermal expansion, which we had not fully incorporated in our initial MD simulation. Our simulations were conducted at a low temperature of 4.2 K, not accounting for thermal expansion effects at higher temperatures. However, according to previous DFT calculations on the mechanistic properties of the TMDs, the predicted changes to the lattice constant due to thermal expansion at 500 K are only $\sim 0.22\%$ and $\sim 0.26\%$ for MoS₂ and MoSe₂, respectively¹. Moreover, this marginal disparity of 0.04% between the two compounds is considered negligible to the conformation of the 2D lateral superlattice obtained in our simulation. Therefore, while thermal expansion is a relevant factor, we believe it does not significantly impact the overall conclusions drawn from our simulations at the chosen temperature. We have added a discussion in the manuscript to clarify this point and provide evidence from the literature to support our assertion.

Inclusion of Substrate:

Moving on to the substrate effect, as the reviewer rightly pointed out, the interaction between the substrate and the two-dimensional (2D) material is a significant factor in the material's deformation. We expect that the conformation of ripple/buckle in the TMD superlattice is affected by two main governing factors, the lattice mismatch in the lateral superlattice, and the vdW force between the TMD and the substrate. Indeed, we have observed that buckles instead of ripples tend to form on the wide width MoSe₂ regions and that the ripple height is relatively small in narrow width MoSe₂ regions, where the substrate interaction seems to suppress the ripple/buckle formation to some extent (See below **Fig. 3c**, **Supplementary Fig. 1e**, and **Supplementary Fig. 4a**).

Fig. 3c

Supplementary Fig. 1e & 4a

The referenced study by Al-Mulla et al. provides valuable insights into the adhesion energy's role in the deformation of another 2D material, graphene. However, we must note that the authors of the paper introduced interaction between graphene and hypothetical beads with arbitrary interaction strength. Indeed, applying realistic substrate interaction is not a simple task.

To our knowledge, there is no previously fitted MD potential modeling the interaction of lateral TMD superlattice with a real substrate (SiO₂ in our case). Generating such a potential that can accurately account for the van der Waals interaction between the three-component lateral superlattice and the two-component substrate is a non-trivial work and an independent research topic in itself, which requires extensive DFT calculations for potential fitting, a task beyond the scope of the current study. While we could do similar work as the mentioned paper, such work still cannot give a definitive correct modeling of the TMD-SiO₂ system.

Under the circumstance that the substrate is not included, the choice of our simulation parameters, 4.2 K and 100 ps, reduces the deformation and drifting of the free monolayer. The resulting conformation is found to be in better match to the experimental observations in terms of wavelength and amplitude of the ripples in **Fig. 3c** and **Supplementary Fig. 1e** (~ 170 nm, 3 nm in wavelength and amplitude in simulation). Thus, the obtained structures are suitable for further analysis of kMC, in the sense that the conformation is similar to experimental ones. On the other hand, high-temperature simulations of the monolayer lead to significant flexural deformation and drifting, with the ripples being merged into a few big ones, deviating from the experimental observations.

If the substrate and the higher temperature is included together, we expect that we can reproduce the abrupt buckles in wide MoSe₂ region, with flattening of some parts occurring due to vdW interaction to the substrate, and extremely strained buckles occurring to relieve the lattice mismatch within the TMD (depicted in **Supplementary Fig. 4d** and

below). But as we have argued, such a situation is still explainable by the mechanism we have uncovered, with the initial nucleation of ALD still occurring at the concave regions at the start of the buckles.

Supplementary Fig. 4d

All in all, we acknowledge that our current MD-obtained conformation has limitations due to the missing substrate and thermal effects, which means the conformation of the obtained TMD superlattice is different from the real surface. However, we have chosen the simulation parameters that will best reflect the lattice-mismatch-induced rippling. We think that this choice is sufficient to give insights into the trends in the surface conformation and resulting differences in reaction rates, as the existence of strain and curvature itself already affect trends in nucleation positions, albeit some structural differences due to lacking a substrate. In a follow-up study, we plan to further investigate the effect of interfacial interaction between the TMD and the substrate through parametrization of the interlayer potential.

[Action in Manuscript]

i) We have revised the manuscript to make it clearer that our MD simulation consists of a monolayer TMD, and that low temperature was employed to exclude thermal deformation and drifting (p. 19).

(p. 19) To simulate strain arising from lattice mismatch in the lateral superlattice, we construct monolayer lateral MoS₂-MoSe₂ structures. Given that no substrate was included in the simulation, the monolayer was able to deform freely, potentially leading to conformations divergent from experimental observations. In order to minimize such deformations while retaining the effects of lattice mismatch, after energy minimization of the cell with the conjugate gradient method, NPT simulations were performed at a low temperature of 4.2 K and 1 bar pressure in X and Y direction for 100 ps, as had been suggested to exclude the intervention of thermal deformation and drifting^{39,40}. Although the low temperature inevitably excludes the effects of thermal expansion, prior computational studies have indicated that the difference in lattice expansion between MoS₂ and MoSe₂ is negligible (less than 0.04%) even at elevated temperatures of 500 K, thereby implying a minimal impact on strain properties⁴¹. The choice of temperature and timescale resulted in ripples with wavelength in the range of ~170 nm, which is consistent with experimental observations on the 1:1 size matched periodic slab with repetitive stripes of 40 nm and 100 nm MoSe₂ and MoS₂.

(2) The authors perform significant amount of calculations on the adsorption energy of the stress-induced substrates and vacancies. However, the slight difference of the adsorption energy and energy difference cannot interpret the obvious selectivity. The KMC simulation which includes the adsorption, diffusion and desorption of the ALD precursors shows that the TMA has the longest residence and the most active collisions in the MoSe₂ valley region. The authors attribute the selective deposition to combination the adsorption, diffusion and desorption of the ALD precursors. However, the residence time and collisions are indirectly correlated to the selectivity. The coverage of Al species should be provided by the KMC, which can quantitatively represent the selectivity between the two regions.

[Author reply]

We appreciate the opportunity to clarify the aspects of selectivity raised by the reviewer. We acknowledge that direct visualization of Al species, presumed to be Al₂O₃ nucleates, would provide the most direct evidence of selectivity. However, directly demonstrating the formation of Al₂O₃ is beyond our current simulation capabilities due to computational costs and the limitations in the lattice kMC methodology we employed.

Our kMC simulation was designed to elucidate the likelihood of initial nucleation in specific regions, such as the MoSe₂ valley. The input rates for adsorption, desorption, and diffusion of TMA on each chalcogen site reflect the influence of local strain/curvature and the unique environment at each site due to rippling on the TMD. To maintain manageable computational costs, we adopted a simplified model simulating the first TMA pulse. This approach yielded insights into residence times and collision frequencies at various chalcogen sites. While our simulations primarily track the movement of TMAs, we expect that the observed adsorption and diffusion rates are indicative of other Al molecular intermediates like DMAOH, given their structural similarities.

The amorphous nature of Al₂O₃ nucleates and the necessity of continuously defining new reaction sites during growth pose significant computational hurdles. A comprehensive DFT analysis to incorporate the variable energies of TMA+H₂O reactions across different TMD environments, and the diverse reactions between Al intermediates and H₂O, would be extensive. The kMC model would also need updating to accommodate new reaction sites that emerge on the amorphous Al₂O₃, meaning we must implement a 3D kMC with arbitrary site positions to represent additional reaction sites on the Al₂O₃ instead of a 2D lattice kMC, necessitating an independent line of research far from our current scope.

While residence time may be indirectly related to area selectivity, collision frequencies of TMAs are a more direct correlate. The frequency of dimerization reactions, forming two-Al intermediate molecules, should be proportional to collision frequency. These bulkier dimer molecules would exhibit significantly reduced diffusion and desorption rates, fostering further reactions and nucleation. We have observed that some sites located at the Se valley have higher collision frequencies, enhancing nucleation potential—this effect will be amplified with iterative ALD cycles with the nucleation probability being multiplied. Such disparities are crucial for understanding nucleation likelihood and growth patterns during the ALD process.

To show more clearly the population of high collision frequency sites in MoS₂ and MoSe₂ regions, we have reorganized the heatmap in **Supplementary Fig. 18** to a population density plot below in **Fig. R1**. At higher temperatures where the overall coverage is lower, the high collision region becomes more pronounced forming a shoulder, marked by the red arrows. This more clearly indicates that some regions of the MoSe₂ possess high collision sites at the Se valley, and is in line with the observation that higher temperatures lead to better area selectivity, supporting our claims.

Fig. R1 | Normalized collision probability density plot for kMC simulation of 383 (a), 413 (b), 443 (c), and 473 K (d). The x-axis represents the collision count divided by the average number of collisions, while the y-axis represents the probability density. Red arrow marks the population of high collision sites apparent in MoSe₂ at 443, 473 K simulations.

[Action in Manuscript]

i) We have revised the manuscript and Supplementary Information Note 7 to more clearly indicate the simulation method implemented, and emphasize the importance of initial TMA collision in the nucleation position. (p.12 & S.I. p. 33, 36)

(p. 12) Since the diffusion of TMA molecules on the TMD surface is very rapid, the collision frequency of TMA can also serve as a good estimate of nucleation in SAS-ALD, given the structural similarities between the TMAs and the reactive single Al(OH)_n(CH₃)_{3-n} intermediates

(S.I. p. 33) The lattice kMC simulation of TMAs on the TMD surface was carried out with the aim of proving the validity of our mechanism, providing numerical evidence within a real-world context.

(S.I. p. 36) This estimation is based on the expectation that the reactive $\text{Al}(\text{OH})_n(\text{CH}_3)_{3-n}$ intermediates, formed after the initial H_2O pulse, will exhibit diffusion rates comparable to those of the structurally similar TMAs. The dimeric aluminum intermediates, resulting from the collision of these intermediates will possess significantly decreased diffusion and desorption rates compared to the single Al intermediates, primarily due to enhanced van der Waals (vdW) interactions as a result of the increased mass of the intermediate. Consequently, the collision probability and dynamics involving monomeric aluminum intermediates at various lattice sites may provide insights into the likely positions of subsequent nucleation processes.

(3) Although the authors claim that they focus on uncovering the SAS-ALD mechanism, it is still concerned that the interfaces between the ALD film and substrate need to be described. It is an ongoing question for an ideal interface between ALD film and 2D materials. The interface property would significantly alter the electrical properties of the dielectric oxide films. Therefore, the leakage current, breakdown voltage and EOT needed to be explored to support this work for potential application. It is also notice that the authors have provided contact measurement for previous reviewer's concern, and the results show that the contact has poor electrical properties. It would be beneficial for the authors to provide possible discussions and solutions to improve the electrical properties via interface engineering.

[Author reply]

We sincerely appreciate your insightful comment. Due to the technical limitations of lithography and device fabrication, precisely evaluating the electrical properties of our selectively deposited Al_2O_3 , with its features in the tens of nanometers, is challenging. However, to address your concerns and to characterize the properties of the selectively deposited Al_2O_3 , we have added the following experimental results,

- (i) Compositional analysis using cross-sectional TEM HAADF and EDS spatial mapping (Fig. R2);
- (ii) Optical property and band gap analysis via UV-VIS spectrum (Fig. R3);
- (iii) Electrical property evaluation through FET measurements, utilizing Al_2O_3 as the gate insulator on 2D superlattice channel (Fig. R4).

(i) Interface quality between aluminum oxide and MoSe_2 surface

Fig. R2a is a cross-sectional HAADF-STEM image showing the selectively deposited aluminum oxide on the MoSe_2 region. The white line in the image corresponds to the MoS_2 - MoSe_2 surface, and the bright gray region above the white line represents aluminum oxide. We observed that there are no serious defects such as cavities at the interface between MoSe_2 and aluminum oxide, indicating a flat interface with good contact quality. Moreover, the clear image of the MoSe_2 suggests that AS-ALD does not cause degradation to the 2D surfaces. **Fig. R2b** and **R2c** show EDS mapping images of the SAS-ALD structure, representing the distribution of aluminum and oxygen elements, respectively. The selectively deposited aluminum oxide region exhibits a same element signal in both areas, near and far from the interface. Thus, the quality of aluminum oxide at the interface is expected to be identical to that in the bulk. The images observed through STEM indicate a clean and uniform interface between the 2D material and selectively deposited oxide.

Fig. R2 | HAADF-STEM and EDS mapping of aluminum oxide SAS-ALD. a-c Cross-sectional HAADF-STEM (a) and EDS mapping images (b, c) of aluminum oxide SAS-ALD structure on MoS_2 - MoSe_2 superlattice. EDS mapping images show aluminum (b, purple) and oxygen (c, blue) elements.

(ii) Bandgap of aluminum oxide by SAS-ALD

Fig. R3a shows an absorbance Tauc plot measured by UV-VIS spectrophotometer. A monolayer MoS_2 - MoSe_2 lateral superlattice film was grown on the Si/SiO_2 substrate and transferred onto a transparent fused silica substrate. Lastly, aluminum oxide was deposited on the superlattice film by ALD (100 cycles). Typically, an amorphous aluminum oxide has a bandgap of 5 to 7 eV range²⁻⁵. In our case, the bandgap of the aluminum oxide, extracted from the Tauc plot in **Fig. R3a**, is 5.2 eV, which is in valid range of bandgap. **Fig. R3b** is an enlarged absorbance spectrum of the green square region in **Fig. R3a**, revealing that the bandgap of the MoS_2 - MoSe_2 film is 1.66 eV.

Fig. R3 | Bandgap of aluminum oxide in SAS-ALD by UV-VIS spectrophotometer. a An absorption spectrum of aluminum oxide on the MoS₂-MoSe₂ lateral superlattice film by SAS-ALD. The bandgap of aluminum oxide is 5.2 eV in the Tauc plot. **b** The enlarged spectrum of green square region in **a**. The bandgap of MoS₂-MoSe₂ lateral superlattice film is 1.66 eV.

(iii) Top gate FET operation of MoS₂-MoSe₂ channel

We evaluated the gate insulator characteristics of the deposited Al₂O₃ using a MoS₂-MoSe₂ channel for FET fabrication. By extending the growth time, we induced lateral growth of Al₂O₃, ensuring it fully covered the van der Waals surfaces of the entire channel.

Fig. R4a and **R4b** shows a SEM image and schematic of a top gate FET with a MoS₂-MoSe₂ superlattice as a channel. MoS₂-MoSe₂ channel region has 1.2 μm length and 2.1 μm width, and Bi/Au (15/40 nm) was deposited as source and drain electrodes. The top gate insulator (AlO_x) was deposited by the ALD process with a thickness exceeding 30 nm, fully merging it onto the superlattice surface. A gate electrode was Cr/Au (10/40 nm). **Fig. R4c** shows transfer curve of the top gate FET which represents electrical properties. The measurement was conducted in the vacuum chamber (under 20 mTorr) at room temperature. The mobility is about 8 cm²/V·s, the on/off ratio is about 10³, and the subthreshold swing is 2.78 V/dec. Although a direct comparison is challenging due to the lack of existing references for MoSe₂-MoS₂ superlattice channel FETs, we have clearly demonstrated that the deposited aluminum oxide on this superlattice surface is of sufficient quality to function as a gate insulator for FETs.

Fig. R4 | Aluminum oxide top gate FET operation of MoS₂-MoSe₂ channel. a, b SEM image (a) and schematic (b) of MoS₂-MoSe₂ top gate FET. **c** Transfer curve of the aluminum oxide top gate FET measured in vacuum and room temperature condition. The channel region is composed of MoS₂ and MoSe₂ regions. The gate oxide (AlO_x) is fully merged on the superlattice surface with over 30 nm thickness.

[Action in Manuscript]

i) We added Supplementary Fig. 20 (S.I. p. 21), Supplementary Information Note 8 (S.I. p. 37-38), explanation (p. 5), and method (p. 17) about interface quality and optical property of SAS-ALD structure.

(S.I. p. 37-38) We discussed the selective deposition of various materials on the 2D lateral superlattices. To increase a scalability of our method, it is necessary to analyze the characteristics of the selectively deposited materials.

First, we analyzed the interface quality between the deposited materials and the 2D surface. Supplementary Fig. 20a is a cross-sectional HAADF-STEM image showing the selectively deposited aluminum oxide on the MoSe₂ region. The white line in the image corresponds to the MoS₂-MoSe₂ surface, and the bright gray region above the white line represents aluminum oxide. We observed that there are no serious defects such as cavities at the interface between MoSe₂ and aluminum oxide, indicating a flat interface with good contact quality. Moreover, the clear image of the MoSe₂ suggests that AS-ALD does not cause degradation to the 2D surfaces. Supplementary Fig. 20b and 20c show EDS mapping images of the SAS-ALD structure, representing the distribution of aluminum and oxygen elements, respectively.

The selectively deposited aluminum oxide region exhibits a same element signal in both areas, near and far from the interface. Thus, the quality of aluminum oxide at the interface is expected to be identical to that in the bulk. The images observed through STEM indicate a clean and uniform interface between the 2D material and selectively deposited oxide.

Next, we measured the optical properties of the selectively deposited aluminum oxide on the MoSe₂ region. Supplementary Fig. 20d shows an absorbance Tauc plot measured by UV-VIS spectrophotometer. A monolayer MoS₂-MoSe₂ lateral superlattice film was grown on the Si/SiO₂ substrate and transferred onto a transparent fused silica substrate. Lastly, aluminum oxide was deposited on the superlattice film by ALD (100 cycles). Typically, an amorphous aluminum oxide has a bandgap of 5 to 7 eV range. In our case, the bandgap of the aluminum oxide, extracted from the Tauc plot in Supplementary Fig. 20d, is 5.2 eV, which is in valid range of bandgap. Supplementary Fig. 20e is an enlarged absorbance spectrum of the green square region in Supplementary Fig. 20d, revealing that the bandgap of the MoS₂-MoSe₂ film is 1.66 eV. Through the aforementioned analyses, we can confirm that the material deposited by the SAS-ALD technique exhibits a good interface on the 2D surface and demonstrates valid properties.

(p. 5) Additionally, the interface between Al₂O₃ and MoSe₂ is clean, and the aluminum oxide exhibits valid bandgap values (S. I. Note 8).

(p. 17) The bandgap of aluminum oxide and superlattice film were measured by UV-VIS spectrophotometer (Lambda 1050).

ii) We have added comments about importance of confirming and improving AS-ALD materials' properties to the outlook explanation (p. 14).

(p. 14) For instance, TMD nanoribbon device can be easily fabricated with SAS-ALD structures (see S. I. Note 9). Also, when using AS-ALD materials as gate oxide or metal contact for the underlying 2D semiconductors, our unique 2D/3D structures can be implemented in advanced 2D devices such as short-channel device. To achieve this, it is necessary to confirm and improve the characteristics of AS-ALD materials.

(4) It is also noticed that another reviewer has raised concerns on the defects and interface region induced selective deposition. The authors have added reasonable discussions about the influence of different defects to selective growth. The grain boundaries region between two different 2D materials are important areas to impact the selective growth. Although the authors try to decouple the influences of the grain boundary, they shall also discuss clearly the initial nucleation stage on the interface areas and on MoSe₂. In addition, it is suggested to demonstrate the ALD nucleation stage on MoS₂-MoSe₂ substrates with larger MoSe₂ width. All these could help to decouple the grain boundaries region effect as raised up by the previous reviewer.

[Author reply]

As mentioned by the reviewer, the interface region between MoS₂ and MoSe₂ can also affect our selective deposition. While the defect density in the interface region is not significantly higher than in other regions (**Supplementary Fig. 1d**), it is necessary to analyze the initial nucleation in the interface as some strains can be induced, as shown in **Fig. 3d**.

Supplementary Fig. 1d

Fig. 3d

When the MoSe₂ width is sufficiently narrow (< 50 nm), as shown in **Fig. 3a**, aluminum oxide is periodically deposited similarly to the ripple periodicity in MoSe₂. This indicates that the interface region itself does not significantly impact selective deposition, as the interface area is very close to the MoSe₂ valley sites. As explained in **Fig. 4**, TMA molecules can diffuse on the 2D surfaces, and when the distance between the interface region and the ripple is very close (**Fig. R5a**), most TMA molecules sufficiently move to the MoSe₂ valley region. Therefore, in this case, the interface region can't effectively serve as initial nucleation sites.

Fig. 3a

Fig. 4

Fig. R5 | Initial nucleation sites in small and large MoSe₂ superlattice. a, b Schematics of initial nucleation sites on the MoS₂-MoSe₂ lateral superlattice with small (a) and large (b) MoSe₂ width. The green region around TMA means the available diffusion region of the molecule. In the small MoSe₂ width case (a), aluminum oxide nuclei occur on the MoSe₂ valley region. In the large MoSe₂ width case (b), nuclei occur on the MoSe₂ buckle and interface regions.

However, when the MoSe₂ width becomes wider (> 200 nm), initial nucleation of aluminum oxide can be observed in both the MoSe₂ buckle region and the interface region, as shown in **Supplementary Fig. 4**. It is anticipated that as the distance between the interface and the MoSe₂ buckle region increases, some TMA molecules can't move entirely to the buckle area due to the long distance. This insufficient diffusion causes another initial nucleation site, the interface region between MoS₂ and MoSe₂, which is less stable than the MoSe₂ buckle but more stable than the MoS₂ region (**Fig. R5b**). In other words, if the distance for the ALD precursor to undergo surface diffusion to the most stable region (compressed MoSe₂ buckle) becomes too far, some initial nucleation can also occur in the surrounding reasonably stable areas (interface region).

Supplementary Fig. 4

In SAS-ALD, the region most preferred for deposition is the compressed MoSe₂ region. However, if the MoSe₂ valley region is distant from the MoS₂-MoSe₂ interface, preventing some TMA molecules from reaching the MoSe₂ valley through surface diffusion, the next preferred region, the interface area, can undergo initial nucleation. In other words, depending on the superlattice condition, initial nucleation can occur not only in the MoSe₂ valley but also in the interface region.

[Action in Manuscript]

i) We added explanations about the initial nucleation at the MoS₂-MoSe₂ interface region in Supplementary Information Note 3 (S.I. p. 26).

(S.I. p. 26) When the MoSe₂ width is narrow (< 50 nm), Al₂O₃ is primarily deposited in the valley region of the MoSe₂ ripple structure. In contrast, for a larger MoSe₂ width (> 200 nm), initial nucleation occurs not only around the MoSe₂ buckle region but also in the interface regions between MoS₂ and MoSe₂. These two initial nucleation sites will be related to our SAS-ALD mechanism, which exhibits selectivity in adsorption and surface diffusion of the ALD precursors. In the case of a narrow MoSe₂ width, the distance between the MoSe₂ valley region and the interface region is very close (< 50 nm), allowing the ALD precursors to predominantly move to the most stable region, the MoSe₂ valley site. However, as the MoSe₂ width increases, the distance between the MoSe₂ buckle and the interface region widens (> several hundred nm). This can cause the ALD precursors, which are moving around the interface region and not sufficiently reaching the MoSe₂ buckle region, to nucleate not necessarily at the most preferred buckle but rather in a reasonably preferred interface region. In other words, depending on the width of the superlattice, initial nucleation can occur not only on the compressed MoSe₂ region, the energetically most stable sites, but also on the MoS₂-MoSe₂ interface regions.

REVIEWERS' COMMENTS

Reviewer #3 (Remarks to the Author):

In this revision, the authors have provided additional experiments to measure the electrical properties of the dielectric oxide films. The ASD phenomenon on the boundary region are also discussed. From the experiment aspect, the revision is satisfied. Although the authors are not able to address the interactions between the substrate and the TMD layer in their MD simulations, they have provided sufficient discussion and explanation for the associated mechanisms. I have no further comments on the manuscript.